

# The Acid-Catalyzed Hydrolysis of an α-Pinene-Derived Organic Nitrate: Kinetics, Products, Reaction Mechanisms, and Atmospheric Impact

Joel D. Rindelaub, Carlos H. Borca, Matthew A. Hostetler, Mark A. Lipton, Lyudmila V. Slipchenko, Paul B. Shepson

Department of Chemistry, Purdue University, West Lafayette, IN, 47907, USA

*Correspondence to*: Joel D. Rindelaub (jrindela@purdue.edu), Paul B. Shepson (pshepson@purdue.edu)

**Abstract.** The production of atmospheric organic nitrates ($RONO_2$) has a large impact on air quality and climate, due to their contribution to secondary organic aerosol and influence on tropospheric ozone concentrations. Since organic nitrates control the fate of gas phase $NO_x$ ($NO+NO_2$), a byproduct of anthropogenic combustion processes, their atmospheric production and reactivity is of great interest. While the atmospheric reactivity of many relevant organic nitrates is still very uncertain, one significant reactive pathway, condensed phase hydrolysis, has recently been identified as a potential sink for organic nitrate species. The partitioning of gas phase organic nitrates to aerosol particles and subsequent hydrolysis likely removes the oxidized nitrogen from further atmospheric processing, due to large organic nitrate uptake to aerosols and proposed hydrolysis lifetimes, which may impact long range transport of $NO_x$, a tropospheric ozone precursor. Despite the atmospheric importance, the hydrolysis rates and reaction mechanisms for atmospherically-derived organic nitrates are almost completely unknown, including those derived from α-pinene, a biogenic volatile organic compound (BVOC) that is one of the most significant precursors to biogenic secondary organic aerosol (BSOA). To better understand the chemistry that governs the fate of particle phase organic nitrates, this study elucidated the hydrolysis mechanism and rate constants for several organic nitrates, including an α-pinene-derived organic nitrate (APN). A positive trend in hydrolysis rate constants was observed with increasing solution acidity for all organic nitrates studied, with the APN lifetime ranging from 8.3 minutes at acidic pH (0.25) to 8.8 hours at neutral pH (6.9). Since ambient fine aerosol pH values are observed to be acidic, the reported lifetimes, which are much shorter than that of atmospheric fine aerosol, provide important insight into the fate of particle phase organic nitrates. Along with rate constant data, the identification of the products campholenic aldehyde, pinol, and pinocamphone confirms a unimolecular specific acid-catalyzed mechanism is responsible for organic nitrate hydrolysis under acidic conditions, where carbocation rearrangement is favored for α-pinene-derived species. The free energies and enthalpies of the isobutyl nitrate hydrolysis intermediates and products were calculated using a hybrid density functional ($\omega$B97X-V) to support the proposed mechanisms. These findings provide valuable insight into the organic nitrate hydrolysis mechanism and its contribution to the fate of atmospheric $NO_x$, aerosol phase processing, and BSOA composition.





## 1 Introduction

The atmospheric oxidation of biogenic volatile organic compounds (BVOCs), which have annual emission rates (~1100 Tg yr$^{-1}$ total) roughly an order of magnitude larger than anthropogenic non-methane VOCs (Guenther et al., 1995), has a significant impact on air quality and climate. The production of secondary organic aerosol (SOA) from BVOC oxidation products influences the radiative balance of the planet, by directly interacting with both solar and terrestrial radiation, as well as indirectly through their role as cloud condensation nuclei (CCN; e.g. Ramanathan et al., 2001). Overall, the production of SOA from BVOCs has a cooling effect on global climate, estimated to have a combined radiative forcing as large as -1.5 W m$^{-2}$ (Scott et al., 2014). Additionally, the inhalation of SOA has significant impact on the human respiratory system and atmospheric aerosol concentrations are positively correlated with lung cancer and mortality rates (Raaschou-Nielsen et al., 2013). Despite the importance of SOA, the chemical mechanisms that explain the composition of aerosol particles and their chemical processes are still highly uncertain.

The gas phase oxidation of BVOCs also governs tropospheric ozone concentrations by controlling its precursor, NO$_x$ (NO+NO$_2$). In the atmosphere, the most common atmospheric oxidant, the OH radical, can either abstract a hydrogen from or add to a BVOC, if it contains an olefinic functionality (e.g. α-pinene), to create a peroxy radical from the rapid addition of molecular oxygen to the organic radical (Fig. 1). In high NO$_x$ environments, such as areas within atmospheric transport of combustion emissions, nitric oxide can either add to the peroxy radical to form an organic nitrate (RONO$_2$) or it can be oxidized to create an alkoxy radical and NO$_2$, which can readily photolyze to produce ozone (Fig. 1). The ratio of RONO$_2$ production to NO$_2$ production is referred to the organic nitrate branching ratio. Since ozone is a greenhouse gas (IPCC, 2007), damages plants/crops (Fiscus et al., 2005), and is a lung irritant (EPA, 2011), the formation and fate of organic nitrates has implications for both climate and environmental health.

With respect to SOA production, among the most important BVOC-derived organic nitrates are products of α-pinene oxidation, due to their relatively low volatility and the very high annual global emission rate of α-pinene (~66 Tg yr$^{-1}$; Guenther et al., 2012). Under dry conditions, α-pinene-derived organic nitrates (APNs) can comprise a significant fraction of SOA mass (Xu et al., 2015; Rollins et al., 2010). At elevated relative humidity, when aerosol particles have increased liquid water content, organic nitrates can hydrolyze to eliminate the RONO$_2$ functionality (Liu et al., 2013; Rindelaub et al., 2015; Bean and Hildebrandt Ruiz, 2016), leaving the nitrate ion within the particle. However, the products, mechanisms, and kinetics of the α-pinene-derived organic nitrate (APN) hydrolysis reactions are still unknown, negatively impacting our understanding of aerosol phase chemistry and the fate of atmospheric NO$_x$.

The conversion of the organic nitrate functionality to a non-volatile, largely unreactive nitrate ion via a substitution or elimination mechanism would lead to an effective sink of atmospheric NO$_x$ and reduce the potential for NO$_x$/O$_3$ transport. Recent results from Romer et al. (2016) indicate that the lifetime of atmospheric boundary layer NO$_x$ could be as low as ~2 hours, using an assumed short hydrolysis lifetime (Romer et al., 2016). The hydrolysis mechanism could also potentially impact SOA formation and cloud condensation nuclei activity. Thus the hydrolysis of organic nitrates, and the associated



uncertainty, has a significant impact on our understanding of how BVOC-NO$_x$ interactions affect climate, air quality and health.

Despite much study of organic nitrate hydrolysis, the rates and mechanisms at low pH, which is relevant to both ambient (e.g. Guo et al., 2015) and laboratory conditions (Rindelaub et al, 2016), are still very uncertain. While S$_N$2
mechanisms are believed to be more prevalent at high pH (Baker and Easty, 1950; Boschan et al., 1955), recent studies suggest that unimolecular mechanisms are responsible for the fate of organic nitrates under aqueous acidic conditions, due to the polar protic solvent system, water's weak nucleophlicity, and the relatively large observed hydrolysis rates for tertiary organic nitrates (Rindelaub et al., 2015). While the reaction is likely acid-catalyzed, the catalysis mechanism is uncertain as both specific and general catalyzed mechanisms have been proposed (Darer et al., 2011; Jacobs et al., 2014).

To better understand the organic nitrate hydrolysis mechanism and kinetics under acidic conditions and the corresponding impact on atmospheric processes, hydrolysis reactions were performed focusing on the fate of α-pinene-derived organic nitrates (APNs). The hydrolysis rate constants, specific mechanisms, and products were determined for a laboratory-synthesized APN. The hydrolysis of simple alkyl nitrates, isopropyl nitrate (IPN) and isobutyl nitrate (IBN), were also studied to gain insight into the mechanisms of primary and secondary substituted species, and to enable computational
chemistry studies of the mechanism and energetics. The results from this study help improve our understanding of organic nitrate chemistry, the fate of atmospherically-relevant organic nitrates relating to climate and health, and can help explain important mechanisms that impact aerosol phase chemistry.

## 2 Experimental

### 2.1 Materials and methods

Organic nitrate hydrolysis reactions were studied for isopropyl nitrate (Sigma Aldrich, >99%), isobutyl nitrate (Sigma Aldrich, >97%), and a β-hydroxy organic nitrate derived from α-pinene by injecting 10 μL of a given standard into a 100 mL buffered solution that was continuously mixed. The α-pinene-derived nitrate, shown in Fig. 1, was synthesized based on Pinto et al. (2007). Briefly, α-pinene oxide was added to a 1.0 M solution of Bi(NO$_3$)$_3$•5H$_2$O in DCM, and stirred for 1 hour under N$_2$ before purification using flash chromatography with a 20% ethyl acetate in hexane solvent system. Product
identification and purity were assessed using NMR (see below). Aliquots of 5 mL were taken at varying time points from the reaction mixture and extracted with 5 mL of tetrachloroethylene (C$_2$Cl$_4$) before analysis using FTIR for organic nitrate quantification and GC-MS for product identification. FT-IR analysis was accomplished by integrating the ~1640 cm$^{-1}$ asymmetric –NO$_2$ stretch unique to organic nitrates (Nielsen et al., 1995). The reaction solutions used were buffered at 10 mM with either a sulfate, acetate, or phosphate buffer system. Hydrolysis reactions were studied at pH values 0.25, 1.0, 4.0,
and 6.9.

The chemical shifts, peak multiplicity and integration of the APN protons in the $^1$H NMR spectrum, using deuterated chloroform (CDCl$_3$) as a solvent, were as follows: (a) δ 5.6 (triplet, 1H), (b) δ 5.6 (singlet, 1H), (c) δ 2.4 (triplet of a triplet,



1H), (d) δ 2.2 (triplet, 1H), (e) δ 1.9 (doublet of a doublet, 2H), (f) δ 1.8 (singlet, 3H), (g) δ 1.6 (singlet, 3H), (h) δ 1.5 (singlet, 3H), (i) (doublet of a doublet, 2H).

## 2.2 Computational methods

The thermochemical calculations of a set of reactants, intermediates and products involved in the proposed reaction pathways of isobutyl nitrate were explored using Density Functional Theory (DFT; Hohenberg and Kohn, 1964; Kohn and Sham, 1965). The set included water ($H_2O$), hydronium ion ($H_3O^+$), nitric acid ($HNO_3$), isobutyl nitrate (IBN), protonated isobutyl nitrate ($IBHN^+$), isobutyl ion ($IB^+$), *tert*-butyl ion ($TB^+$), isobutylene (2MP), *tert*-butyl alcohol (TBA), and isobutyl alcohol (IBA), see Table 1. The reactions are assumed to run in an acidic environment, such that the hydronium ion is prevalent. First, a systematic torsional conformational search was performed on the structure of each molecule of the set, excepting water and hydronium ion. This procedure was performed in HyperChem (Hyperchem™ Professional 7.51, Hypercube, Inc.) with the Optimized Potentials for Liquid Simulations (OPLS) force field (Jorgensen and Tirado-Rives, 1988; Pranata et al., 1991). A maximum of 8 simultaneous variations was allowed, with angles changing every step by a maximum range of 180° at intervals of 15°. Similar structures were filtered, with an acceptance criterion set to 5 kcal mol$^{-1}$ above the lowest energy conformer. All the following calculations were carried out using the computational chemistry package Q-Chem 4.3 (Shao et al., 2015). Second, the lowest energy conformer was optimized employing the long-range corrected hybrid density functional $\omega$B97X-V (Mardirossian and Head-Gordon, 2014), with the aug-cc-pVTZ basis set (Kendall et al., 1992), and Polarizable Continuum Model (PCM) of implicit aqueous solvent (Truong and Stefanovich, 1995; Barone and Cossie, 1998; Cossi et al., 2003). A high-accuracy grid was employed, as well as extremely tight convergence criteria. Third, frequency calculations were executed on the optimized structures to verify the convergence of the geometry optimizations, and also to determine if the molecule was a stable species or a reaction intermediate. These were run using the same setup described above, plus the inclusion of a thermochemical analysis upon completion of frequency calculations.

## 3 Results

In all experiments, the addition of an organic nitrate standard to aqueous solution resulted in hydrolysis of the organic nitrate functionality, with first order loss rates that increased with solution acidity (Figs. 2, 3). For the secondary α-pinene-derived nitrate (APN), hydrolysis rate constants ranged from 3.2 x 10$^{-5}$ s$^{-1}$ at neutral pH (6.9) to 2.0 x 10$^{-3}$ s$^{-1}$ at low pH (0.25). The hydrolysis rate constants for the secondary isopropyl nitrate and the primary isobutyl nitrate displayed nearly identical kinetics, and had rate constants smaller by more than two orders of magnitude relative to the APN, ranging from 1.23 x 10$^{-7}$ s$^{-1}$ at neutral pH (6.9) to 1.1 x 10$^{-5}$ s$^{-1}$ at low pH (0.25), when data from both experiments were averaged together.

The corresponding hydrolysis lifetimes for the organic nitrates studied are shown in Table 2. APN had a condensed phase hydrolysis lifetime of 8.3 minutes at low pH, and a lifetime of 8.8 hours at neutral pH. Both of these hydrolysis lifetimes are much shorter than the lifetime of a typical atmospheric aerosol particle. The average hydrolysis lifetimes of



isopropyl nitrate and isobutyl nitrate were much larger than those for APN, ranging from approximately 1 day at low pH to greater than 8 months at neutral conditions.

The pH dependence of the observed rate constants indicates that the hydrolysis of organic nitrates at low pH is a specific acid-catalyzed mechanism. In specific acid-catalyzed mechanisms, the transfer of the H[+] ion from the acid to the reactant is reversible and occurs before the rate determining step, consistent with a unimolecular mechanism. The observed specific acid-catalyzed mechanism is in contrast to Jacobs et al. (2014), who report a general acid-catalyzed mechanism for the hydrolysis of β-hydroxy organic nitrates. It is important to note, however, that Jacobs et al. (2014) did not report the pH of their solutions, thus, a pH-dependent reaction may have been possible, given their experimental parameters.

Previous studies indicate that organic nitrate hydrolysis rates increase with alkyl substitution (Darer et al., 2011; Hu et al., 2011). However, in this study, essentially identical kinetics were observed for the primary isobutyl nitrate and secondary isopropyl nitrate. This similarity can be explained through inspection of the unimolecular hydrolysis mechanism. A proposed reaction mechanism for the acid-catalyzed hydrolysis of IBN is shown in Fig. 4, where rearrangement to a relatively stable *tert*-butyl carbocation drives the rate of the reaction. A similar observation concerning the relatively large hydrolysis rate constant of a primary organic nitrate was recently reported by Jacobs et al. (2014), who concluded that the resonance stabilization of a primary carbocation increased the rate of a nucleophilic substitution reaction.

To further support the unimolecular reaction mechanism of organic nitrate hydrolysis, theoretical enthalpy and free energy profiles of the proposed isobutyl nitrate reaction mechanism are presented in Fig. 5, (a) and (b), respectively. Based on extensive benchmarks, thermochemical calculations in the gas phase at the $\omega$B97X-V/aug-cc-pVTZ level of theory are accurate up to ~3.6 kcal mol$^{-1}$ (Mardirossian and Head-Gordon, 2014). According to recent literature, the $\omega$B97X-V/aug-cc-pVTZ level of theory offers an excellent balance between computational cost and accuracy (Chan and Radom, 2011; Chan and Radom, 2012). Therefore, the main source of potential inaccuracies in our calculations is the use of a PCM implicit solvent, which is known to provide a less rigorous description of charged species (Takano and Houk, 2005). However, the uncertainty due to using PCM is not quantifiable without calculations involving an explicit solvent model, the pursuit of which is beyond the scope of the present study.

According to the DFT calculations, the isobutyl ion (IB$^+$) corresponds to a saddle point of the energy profile, thus it is considered a metastable reaction intermediate, rather than a stable species. Due to the instability of the primary isobutyl carbocation (IB$^+$), it is likely that a 1,2 hydride shift occurs in concert with bond cleavage of the nitrate group to create the tertiary carbocation intermediate (TB$^+$). In addition, geometry optimizations and frequency calculations indicated that protonation of isobutyl nitrate (IBN) occurs on the terminal oxygen of the nitrate rather than the oxygen of the nitrate ester, as shown in Fig. 4, because the latter produces a metastable species.

Comparing the enthalpy and free energy results, it is observed that both the zero-point vibrational energy and entropic contributions play important roles in determining the most probable products.

Without those contributions a barrier to reach the isobutyl ion is significantly higher and the overall reaction would be much slower. The entropic contribution also impacts the probability of producing isobutylene via an elimination mechanism,





among other products. In any case, computations suggest that the energetically favored product is the nucleophilic substitution product, *tert*-butyl alcohol (TBA). The difference in calculated free energy of the TB$^+$ intermediate and the final products is likely within the uncertainty of using an implicit solvation model for charged species.

The much larger observed hydrolysis rate constants of APN, as well as that for a secondary β-hydroxynitrate from the Jacobs et al. (2014) study, compared to the IPN/IBN systems is related to carbocation stability in the unimolecular mechanism. In addition to greater charge stabilization from its relative size, the α-pinene-derived carbocation can rearrange to form a stable tertiary carbocation, as shown in Fig. 6. Further reaction of the tertiary carbocation intermediate readily occurs, as indicated from product identification using GC-MS. Major products identified from APN hydrolysis were campholenic aldehyde, pinol, and pinocamphone, all of which are derived from the tertiary carbocation (Fig. 6). Thus, the creation of these products from APN hydrolysis further confirms the unimolecular nature of the organic nitrate hydrolysis under acidic conditions. While theoretical calculations were not conducted for this system, the experimental data and supporting theoretical calculations of IBN hydrolysis indicate that the unimolecular mechanism is favored for organic nitrates in acidic environments.

Once the α-pinene-derived tertiary carbocation is formed, the reaction will either proceed via an elimination (E1) mechanism or intramolecular rearrangement (Fig. 6). Following the E1 pathway will lead to the formation of pinocamphone. In this mechanism, water will abstract a β-proton from the tertiary carbocation intermediate, forming a double bond. The resulting olefinic alcohol product will be in equilibrium with pinocamphone through keto-enol tautomerization.

Rearrangement of the α-pinene-derived tertiary carbocation will lead to the formation of either pinol or campholenic aldehyde (Fig. 6). Pinol is formed after the four-membered ring of the 3$^o$ α-pinene-derived carbocation fragments to form a double bond and another tertiary carbocation. This rearrangement will be followed by intramolecular attack from the secondary hydroxyl group to create a protonated pinol compound. The abstraction of the proton by water will complete the acid-catalyzed reaction to create the final pinol product and H$_3$O$^+$.

The major product of APN, campholenic aldehyde, which accounted for over 90% of the total peak area from all products, is formed by rearrangement of the α-pinene-derived tertiary carbocation intermediate to form a secondary carbocation (Fig. 6). The conversion of a tertiary carbocation to a secondary carbocation is usually uphill by about ~10 kcal mol$^{-1}$, however, if the rearrangement leads to the formation of a product much lower in energy, this barrier is not prohibitive (Carey and Sunberg, 2007). In this case, the rearrangement of the less stable bicyclo[3.1.1]heptane system to the more stable bicyclo[2.2.1]heptane system compensates for the energetic difference between secondary and tertiary carbocations. After rearrangement to form the secondary carbocation, fragmentation occurs via a retro-Prins reaction to create a cyclopentene and an aldehyde. The final product, campholenic aldehyde, is formed after water abstracts the remaining acidic proton, reforming H$_3$O$^+$ and completing the acid-catalyzed reaction.



## 4 Discussion

As discussed above, particle phase and cloud water hydrolysis is an important reaction concerning the atmospheric fate of organic nitrates. The consumption of the $RONO_2$ functional group within the aerosol phase has an impact on the fate of atmospheric $NO_x$ and its contribution to ozone concentrations. The rapid conversion of the $RONO_2$ functional group to the nitrate ion, which has negligible vapor pressure and will exist within particles depending on pH, indicates that the partitioning of atmospheric organic nitrates to aerosol particles is not likely to induce further reactions capable of re-releasing gas phase $NO_x$ to the atmosphere, which will greatly diminish the potential for long range transport of $NO_x$ in the form of organic nitrates.

Additionally, with recently reported aerosol pH values ranging from pH 0.5 to 3.0 in the southeastern US (Guo et al., 2015), the corresponding ambient hydrolysis lifetimes of APNs would be on the order of a half hour, further indicating that particle phase hydrolysis is a likely efficient sink for atmospheric organic nitrate compounds. Hydrolysis in chamber experiments may be even faster as aerosol pH has recently been measured as low as pH -0.68 for laboratory-generated particles (Rindelaub et al., 2016). In addition, many other α-pinene and monoterpene-derived derived organic nitrates are expected to be tertiary (Peeters et al., 2001), which are likely more reactive than the APN studied within these experiments and will have larger hydrolysis rates. This also indicates that the current ambient measurements of particle phase organic nitrate concentrations may be underestimating the atmospheric production of organic nitrates, due to the likely large degree of aerosol phase hydrolysis. Indeed, this chemistry can represent a dominant fate of $NO_x$ in forested boundary layers and, at low aerosol pH, protonation of the resultant $NO_3^-$ can represent a dominant source of atmospheric $HNO_3$ (Romer et al., 2016).

The formation of relatively high vapor pressure products from α-pinene-derived nitrate hydrolysis, such as campholenic aldehyde, may lead to a reduction in aerosol mass by the partitioning of products back into the gas phase, lowering particle mass concentrations. For instance, the calculated vapor pressure of campholenic aldehyde is estimated to be three orders of magnitude greater than the original organic nitrate, based on calculations using the EPI Suite available at the Environmental Protection Agency website (http://www.epa.gov/opptintr/exposure/pubs/episuite.htm). It is important to note that APN hydrolysis products can have olefinic functionality, such as the case with campholenic aldehyde and pinol, and may react further in the particle phase, especially under acidic conditions, where sulfanation and/or oligomerization can occur. Photo-induced chemistry occurring to aerosol phase products may result in oxidation at the double bond (Bateman et al., 2011). Campholenic aldehyde has also been identified as the major product of the hydrolysis of another α-pinene oxidation product, α-pinene oxide (Bleier and Elrod, 2013), thus, campholenic aldehyde may be an important tracer for the hydrolysis of α-pinene-derived species. Both the gas and particle phase fate of campholenic aldehyde warrant further study.

The identification of organic nitrate hydrolysis is important not only to our understanding of the atmosphere but also to our chemical understanding of organic nitrate hydrolysis. Research regarding $RONO_2$ hydrolysis under acidic conditions has been limited and suggests that nucleophilic substitution is the dominant reaction pathway. This study shows that through the



unimolecular mechanism, elimination and intramolecular rearrangement are also likely reactive pathways and should be considered when identifying aerosol phase chemical processes and potential tracers of atmospherically relevant compounds.

## 5 Conclusions

A specific acid-catalyzed hydrolysis mechanism was determined for an α-pinene-derived organic nitrate, which has implications into atmospheric air quality and climate. This finding, along with supporting theoretical calculations of the isobutyl nitrate hydrolysis mechanism, helps broaden our chemical understanding of the hydrolysis mechanism of organic nitrates. The hydrolysis rates observed for the organic nitrates studied increased with solution acidity and the large rates observed for the α-pinene-derived organic nitrate further emphasizes the likelihood of particle phase hydrolysis being a sink for organic nitrates and, transitively, atmospheric $NO_x$. It also highlights the importance of ambient aerosol pH measurements. The hydrolysis of organic nitrates within the particle phase will lead to a decreased effective lifetime for $NO_x$ and, thus, decreased ozone transport. However, some of the organic hydrolysis products are relatively volatile and may partition back to the gas phase, decreasing organic aerosol mass. Future work is needed to assess how the loss of particle phase organic nitrates impacts cloud condensation nuclei and environmental health.

*Acknowledgements.* This research was supported in part through computational resources provided by Information Technology at Purdue University. P.B.S. acknowledges support from the National Science Foundation (grant AGS-1228496) and L.V.S. acknowledges support from the National Science Foundation (grants CHE-1465154).

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





**Table 1: Stoichiometry of the calculated states.**

| State | Species Included |
|-------|------------------|
| IBN | $(CH_3)_2CHCH_2ONO_2 + H_3O^+ + H_2O$ |
| IBHN$^+$ | $(CH_3)_2CHCH_2ONO_2H^+ + 2H_2O$ |
| IB$^+$ | $(CH_3)_2CHCH_2^+ + HNO_3 + 2H_2O$ |
| TB$^+$ | $(CH_3)_2C^+CH_3 + HNO_3 + 2H_2O$ |
| 2MP | $(CH_3)_2C=CH_2 + HNO_3 + H_3O^+ + H_2O$ |
| IBA | $(CH_3)_2CHCH_2OH + HNO_3 + H_3O^+$ |
| TBA | $(CH_3)_3COH + HNO_3 + H_3O^+$ |





**Table 2: The hydrolysis lifetimes of isopropyl nitrate (IPN), isobutyl nitrate (IBN), and the α-pinene-derived nitrate (APN) at varying pH.**

| pH | Lifetime | | |
|---|---|---|---|
| | IPN | IBN | APN |
| 0.25 | 28 h | 23 h | 8.3 min |
| 1.0 | -- | -- | 44 min |
| 1.7 | 35 d | 33 d | -- |
| 2.5 | -- | -- | 33 min |
| 4.0 | 30 d | 28 d | 1.3 h |
| 6.9 | > 8 mo | > 8 mo | 8.8 h |


**Figure 1: The formation of atmospheric organic nitrates and ozone from the gas phase oxidation of α-pinene, initiated by the OH radical.**



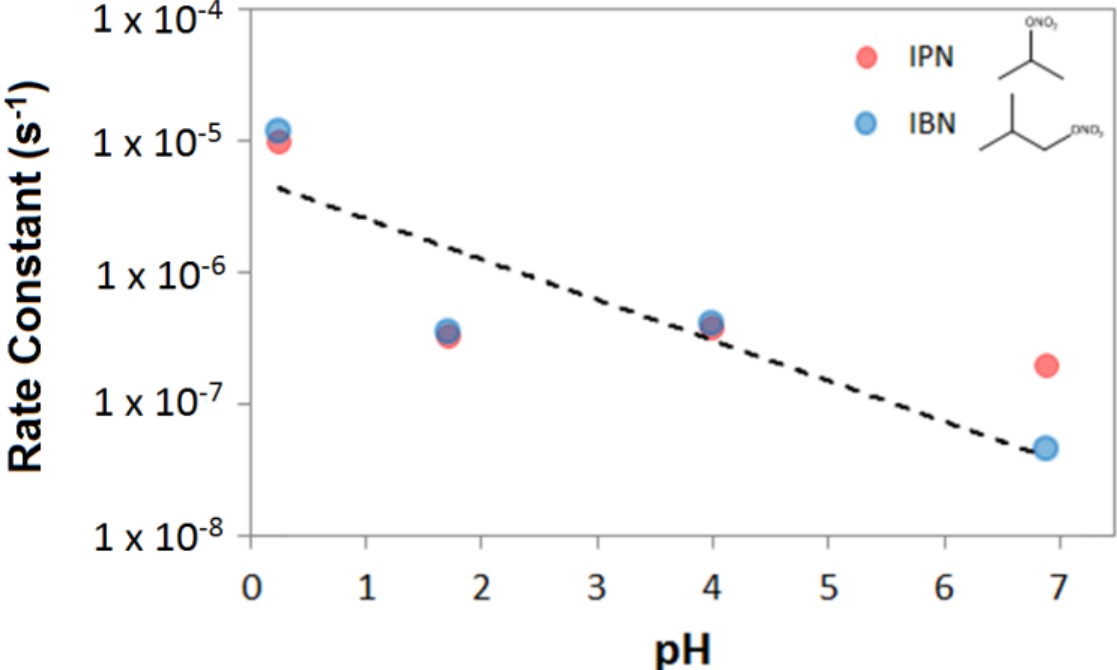

**Figure 2: The hydrolysis rate constants (s⁻¹) for isopropyl nitrate (IPN; red) and isobutyl nitrate (IBN; blue) as a function of solution pH.**





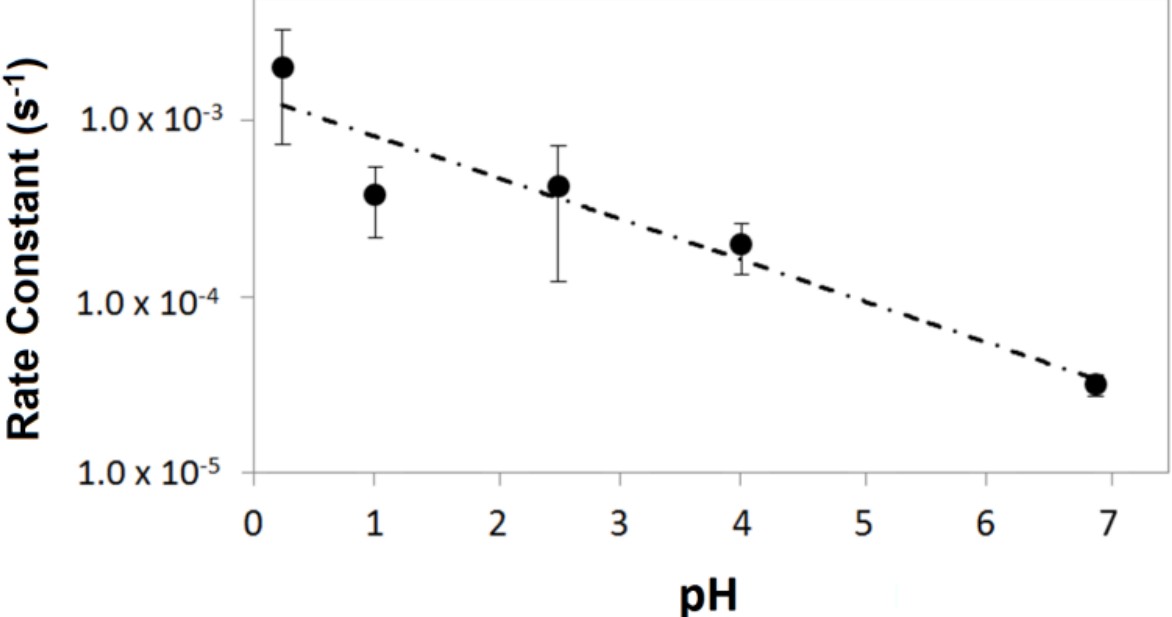

**Figure 3: The hydrolysis rate constants (s$^{-1}$) for the α-pinene-derived nitrate as a function of solution pH. The error bars correspond to one standard deviation of replicate measurements.**




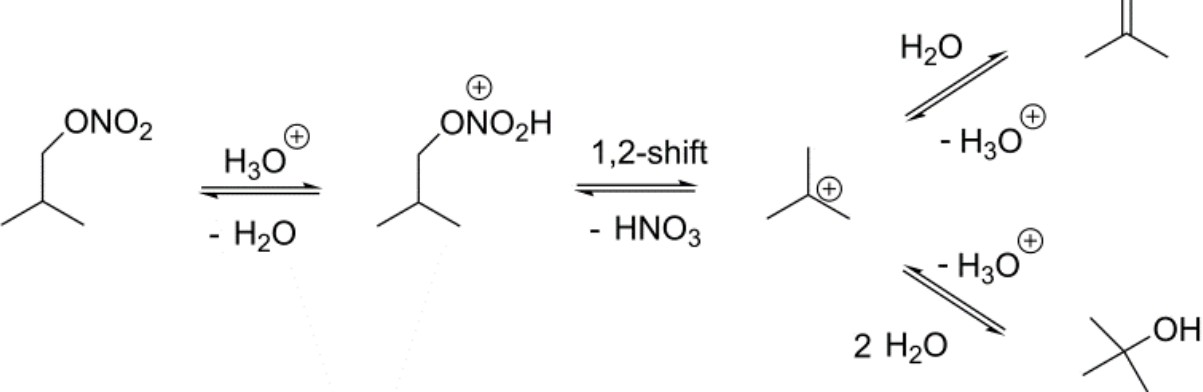

**Figure 4: The proposed unimolecular mechanism for isobutyl nitrate (IBN) demonstrating the specific acid-catalysis and 1,2 hydride shift rearrangement.**





**Figure 5: The calculated relative free energies of the intermediates and products of the proposed acid-catalyzed isobutyl nitrate hydrolysis mechanism. See Table 1 for calculation stoichiometry. The reaction is not likely to proceed through the IB$^+$ intermediate, due to the instability of the carbocation.**



**Figure 6. The proposed acid-catalyzed hydrolysis mechanism of an α-pinene-derived nitrate.**