# Peer review of "The Acid-Catalyzed Hydrolysis of an $\alpha$ -Pinene-Derived Organic Nitrate: Kinetics, Products, Reaction Mechanisms, and Atmospheric Impact"

_Atmospheric Chemistry and Physics, 2016_

## Referee Comment (RC1) · Anonymous Referee #1 · 29 Aug 2016

This manuscript describes bulk solution experiments aimed at understanding the possible aerosol phase hydrolysis of organic nitrates derived from alpha-pinene under conditions of varying acidity. The main finding of the work is that the kinetics of the hydrolysis process is fast compared to aerosol lifetimes, and is acid-catalyzed, indicating that the process will be even faster for low pH aerosol particles. This finding would be important in that it suggests that aerosol phase partitioning of organic nitrates may be an important sink for NOx, with a correspondingly important impact on the accuracy of atmospheric chemical models.

However, there are two aspects that lead me to be concerned that the authors have not definitively identified the chemical species that they have studied.

[Figure]

One aspect is a problem with the manuscript itself. While I think that that the authors are claiming that they have synthesized and studied the secondary organonitrate depicted in Figure 6 from hints in the text discussion, there is really no discussion of how the authors determined which of the isomers they determined they had actually synthesized, and Figure 1 confusingly shows the tertiary organonitrate.

The other aspect is more fundamental. I don't believe that the 1H NMR chemical shifts reported for the organonitrate species are obviously consistent with either of the two possible isomers. In particular, two chemically inequivalent protons with chemical shifts of about 5.6 ppm are reported. Muthuramu et al. ES&T 1993, 27, 1117-1124 demonstrated that protons that are geminal to nitrate groups for simple hydroxynitrates shift about 1.2 ppm higher compared to protons that are bonded to carbons with a germinal hydroxyl group instead (for example, compare the relevant proton in 2-nitrooxy-3-hydroxybutane to 2,3-butanediol), while the hydroxyl protons themselves shift to a lower chemical shift value in the hydroxynitrates as compared to the diols. This information, in conjunction with the NMR spectrum for pinanediol, can be used to predict the proton chemical shifts for the proposed organonitrates. Zhu et al. J. Med. Chem. 2009, 52, 4192–4199 reported the 1H NMR spectrum for the two stereoisomers of pinanediol. For both stereoisomers, the proton on the secondary carbon bonded to a hydroxyl group is observed at about 4.0 ppm. The hydroxyl group protons are observed, depending on the particular stereoisomer, between 2.8 and 3.5 ppm. All of the other protons are observed at chemical shifts of 2.5 ppm or less. Therefore, for the proposed secondary organonitrate structure, one would expect a single proton chemical shift at about 5.2 ppm (4.0 + 1.2 ppm), which is reasonably consistent with one of the 5.6 ppm peaks reported in the present work. However, no other peaks above about 3.5 ppm would be expected for the secondary organonitrate species. For the tertiary organonitrate species, no peaks at all in the 5-6 ppm range would be expected. On the other hand, chemical shifts in the 5-6 ppm range have been observed for a number of ring-opened olefinic reaction products of alpha-pinene oxide (Bleier et al. JPCA, 2013, 117, 4223-4232), the reagent used to prepare the reactant in the present work.

Therefore, I think the presence of the second 5.6 ppm peak opens up the possibility that the authors have prepared an organonitrate species with a double bond, and it is the reaction of this species that they have actually studied. Of course, it is possible that the proposed secondary organonitrate species has a very unusual 1H spectrum that can't be predicted by the process described above, and the authors have in fact correctly identified the reactant structure (perhaps the authors have additional unreported information, such as a 1H spectrum taken in D2O that indicates that one of the 5.6 ppm peaks is the hydroxyl proton with an anomalously large chemical shift value?). In any case, there certainly isn't enough data reported in the manuscript for such an assignment to be considered definitive. It is probably the case that one dimensional 13C and perhaps 1H-1H and/or 1H-13C two dimensional NMR spectra are needed to definitively identify the structure of the synthesis product.

Additionally, the authors report that campholenic aldehyde, pinol, and pinocamphone reaction products were identified via GC-MS methods. Since none of these species are commercially available as standards, it's not clear to me how definitive identification of the reaction products was made.

Because neither the reactant nor the products of the main chemical reaction under study have been convincingly identified, I believe that these issues must be addressed in a revised manuscript.
* * *

---

## Author Comment (AC1) · 7 Oct 2016

Thank you for the insightful comment. After review of the available NMR spectra, it is clear that we have synthesized an olefinic hydroxyl nitrate, as the reviewer has suggested (see Fig. 1). In addition, a typo was discovered in the reported proton NMR data. The second chemical shift at 5.6 ppm should instead read 4.1 ppm. The manuscript will be revised to incorporate these changes and supplemental information will be added to describe the molecular assignments of both the reactant and products.

[Figure]

[Figure]

**Fig. 1.** The structure of the synthesized alpha-pinene-derived hydroxyl nitrate.

---

## Referee Comment (RC2) · Anonymous Referee #2 · 14 Oct 2016

This work investigated the hydrolysis of a-pinene derived organic nitrate (APN), iso-propyl nitrate, and isobutyl nitrate in solutions of varying pH. It was found that the hydrolysis rate constant increases with solution acidity for all compounds studied. The lifetime of APN with respect to hydrolysis increases from 8.3 min to 8.8 hr for solutions with pH = 0.25 and pH = 6.9, respectively. Campholenic aldehyde, pinol, and pinocamphone were identified as hydrolysis products. Theoretical calculations were performed for the hydrolysis mechanisms for isobutyl nitrate using Density Function Theory, and analogous mechanisms were then proposed for the hydrolysis of APN. A unimolecular specific acid-catalyzed mechanism was proposed.

The manuscript is generally well-written. The work will be of interest to the atmospheric

research community and will add to the growing recent studies on organic nitrate formation and fates. The results are interesting and have important implications. I have several main comments:

Firstly, the manuscript should provide more context and carefully compare and contrast the results from this study to prior published work. Instead of just briefly mentioning prior literature, the authors should discuss any discrepancies in more detail. For instance, with Jacobs et al. (ACP, 2014) with respect to the specific acid-catalyzed mechanism, with Darer et al. (ES&T, 2011) and Hu et al. (ACP, 2011) with respect to the hydrolysis rates of primary/secondary organic nitrates, etc. Further, the results reported here are quite similar to the work by Bleier and Elrod ( J Phys Chem A, 2013), who investigated the hydrolysis of a-pinene oxide under acidic conditions. However, other than mentioning that campholenic aldehyde is also identified as a major product in Bleier and Elrod, there is no further discussion regarding similarities/differences between their results and those reported in Bleier and Elrod. This should be addressed. Bleier and Elrod have also identified other products (pinol, etc) and proposed reaction mechanisms for the formation of these products. In this regard, it'd be important for the authors to specify clearly what is new in this study.

Secondly, the authors did not provide specific details on how the hydrolysis products are identified and verified. Since this is an essential part of the manuscript and affects all the discussions on the proposed mechanisms, this info needs to be included in the revised manuscript.

Lastly, the organics nitrates discussed in this manuscript are from photooxidation under high NOx condition; however, nitrate radical oxidation of BVOC is another important source of organic nitrates (Ng et al., ACPD, 2016), where the relative abundance of primary/secondary/tertiary organic nitrates and their fates can be different from those formed from OH oxidation. When discussing "hydrolysis of organic nitrates" throughout the manuscript, it is important that the authors make it clear regarding the specific origins of the organic nitrates in relevant discussions in the manuscript, and comment

that the results can be different for other types of organic nitrates (from NO3 chemistry).

I recommend publication of the manuscript after the authors address the comments.

Detailed Comments

1. Page 2, line 23. I do not think the authors need to specify "under dry conditions", since organic nitrates are typically detected and can comprise a large fraction of ambient SOA mass even under humid ambient and lab conditions.

2. Page 2, line 26. For hydrolysis of organic nitrates, here, it would be appropriate to cite Darer et al., (2011), Hu et al., (2011), and Boyd et al., (2015) as well.

3. Page 5, line 8. The specific acid-catalyzed mechanism proposed here is in contrast to that in Jacobs et al. The authors noted that in Jacobs et al. the solution pH was not reported. The authors are correct that the pH values were not reported, however, according Jacobs et al, "The hydrolysis of the hydroxynitrate isomers was monitored in a variety of acid concentrations (0 to 2 M HClO4), and, regardless of the acid concentration, the rate of hydrolysis remained constant". With this, it would seem that the range of pH values in Jacobs et al. is comparable to this study? If so, the authors need to evaluate and discuss this discrepancy more extensively.

4. Page 5, line 9. The authors noted that essentially identical kinetics were observed for primary and secondary nitrates, which in in contrast to Darer et al. and Hu et al. where hydrolysis rates increase with alkyl substitution. What is the cause of this discrepancy? Because of different mechanisms? Please discuss in more detail.

5. Page 6, line 8. The authors noted that campholenic aldehyde, pinol, and pinocamphone were identified as major products. a. How is product identification verified? Did the authors synthesize authentic compounds of these products? Please clarify. b. As samples at different time points were analyzed (page 3 line 25), please include time dependent data for these products in the revised manuscript, which can offer insights into the dynamics of the formation of these species and/or potential further reactions

in the particle phase.

6. Page 7, first paragraph. The authors wrote "...partitioning of atmospheric organic nitrates to aerosol particles is not likely to induce further reactions capable of releasing gas phase NOx to the atmosphere...." Other than hydrolysis, organic nitrates can also be photolyzed and/or react with OH further, and depending on the VOC, they can also release NOx back to the atmosphere upon photooxidation (e.g., Nah et al., ES&T, 2016). The authors should modify the text and take both hydrolysis and photooxidation into account when discussing organic nitrates fates and implications of their work.

7. Page 7, line 8. The author wrote "...Campholenic aldehyde has also been identified as the major product of the hydrolysis of another $\alpha$-pinene oxidation product, $\alpha$-pinene oxide (Bleier and Elrod, 2013).".".

a. Other than campholenic aldehyde, just like this study, Bleier and Elrod also identified pinol as a product in their study. Bleier and Elrod also proposed reaction mechanisms for the formation of campholenic aldehyde and pinol. The authors should also mention these results from Bleier and Elrod in their manuscript. b. Bleier and Elrod identified trans-carveol and trans-sobrerol as hydrolysis products in their study. It did not appear that these products are identified in this study? What is the cause of this discrepancy? The experiments in Bleier and Elrod were also conducted under acidic conditions. c. Overall, as this work is quite similar to that by Bleier and Elrod, the authors should compare and contrast their results to those from Bleier and Elrod in more detail, e.g., products, formation mechanisms, etc, and specifically clarify what is new in this study.

8. Page 7, line 13, I think the authors should make it clear that this is the case of organic nitrates formed from photooxidation. For organic nitrates formed from nitrate radical oxidation, the relative amount of primary/secondary/tertiary nitrates and their overall hydrolysis could be different (Boyd et al., ACP, 2015).

9. Figure 1 and Figure 6. Which organic nitrate did the authors synthesize?

10. Figure 6. The chemical structure for pinol is incorrect.

---

## Author Response (AR1)

Dr. Ammann,

In this document, we present, in the order the comments were raised by the reviewers, the changes made to the manuscript and/or our repliers to the reviewers. We believe that we have satisfactorily and completely responded to each and all reviewer issues and that the paper is now in a significantly improved form. We believe that the results provided in this paper represent a sizable step forward in atmospheric chemistry and hope you agree that it is now in a publishable form.

Sincerely,
Joel Rindelaub

Author comments are in **bold**.

Anonymous Referee #1

This manuscript describes bulk solution experiments aimed at understanding the possible aerosol phase hydrolysis of organic nitrates derived from alpha-pinene under conditions of varying acidity. The main finding of the work is that the kinetics of the hydrolysis process is fast compared to aerosol lifetimes, and is acid-catalyzed, indicating that the process will be even faster for low pH aerosol particles. This finding would be important in that it suggests that aerosol phase partitioning of organic nitrates may be an important sink for NOx, with a correspondingly important impact on the accuracy of atmospheric chemical models.

However, there are two aspects that lead me to be concerned that the authors have not definitively identified the chemical species that they have studied.

One aspect is a problem with the manuscript itself. While I think that that the authors are claiming that they have synthesized and studied the secondary organonitrate depicted in Figure 6 from hints in the text discussion, there is really no discussion of how the authors determined which of the isomers they determined they had actually synthesized, and Figure 1 confusingly shows the tertiary organonitrate.

**Thank you for the comment. Figures 1 and 6 have been updated to include the correct structure.**

The other aspect is more fundamental. I don't believe that the 1H NMR chemical shifts reported for the organonitrate species are obviously consistent with either of the two possible isomers. In particular, two chemically inequivalent protons with chemical shifts of about 5.6 ppm are reported. Muthuramu et al. ES&T 1993, 27, 1117-1124 demonstrated that protons that are geminal to nitrate groups for simple hydroxynitrates shift about 1.2 ppm higher compared to protons that are bonded to carbons with a germinal hydroxyl group instead (for example, compare the relevant proton in 2-nitrooxy- 3-hydroxybutane to 2,3-butanediol), while the hydroxyl protons themselves shift to a lower chemical shift value in the hydroxynitrates as compared to the diols. This information, in conjunction with the NMR spectrum for pinanediol, can be used to predict the proton chemical shifts for the proposed organonitrates. Zhu et al. J. Med. Chem. 2009, 52, 4192–4199 reported the 1H NMR spectrum for the two stereoisomers of pinanediol. For both stereoisomers, the proton on the secondary carbon bonded to a hydroxyl group is observed at about 4.0 ppm. The hydroxyl group protons are observed, depending on the particular stereoisomer, between 2.8 and 3.5 ppm. All of the other protons are observed at chemical shifts of 2.5 ppm or less. Therefore, for the proposed secondary organonitrate structure, one would expect a single proton chemical shift at about 5.2 ppm (4.0 + 1.2 ppm), which is reasonably consistent with one of the 5.6 ppm peaks reported in the present work. However, no other peaks above about 3.5 ppm would be expected for the secondary organonitrate species. For the tertiary organonitrate species, no peaks at all in the 5-6 ppm range would be expected. On the other hand, chemical shifts in the 5-6 ppm range have been observed for a number of ring-opened olefinic reaction products of alpha-pinene oxide (Bleier et al. JPCA, 2013, 117, 4223-4232), the reagent used to prepare the reactant in the present work.

Therefore, I think the presence of the second 5.6 ppm peak opens up the possibility that the authors have prepared an organonitrate species with a double bond, and it is the reaction of this species that they have actually studied. Of course, it is possible that the proposed secondary organonitrate species has a very unusual 1H spectrum that can't be predicted by the process described above, and the authors have in fact correctly identified the reactant structure (perhaps the authors have additional unreported information, such as a 1H spectrum taken in D2O that indicates that one of the 5.6 ppm peaks is the hydroxyl proton with an anomalously large chemical shift value?). In any case, there certainly isn't enough data reported in the manuscript for such an assignment to be considered definitive. It is probably the case that one dimensional 13C and perhaps 1H-1H and/or 1H-13C two dimensional NMR spectra are needed to definitively identify the structure of the synthesis product.

**Thank you for the very insightful comment. Supplemental Information (Section S1) was added describing the molecular assignment of the synthesized α-pinene-derived organic nitrate (APN) using $^1$H NMR, $^{13}$C NMR, FTIR, and chemical ionization mass spectrometry (CIMS).**

**It was determined, as the reviewer suggested, that the APN has olefinic functionality, thanks to supporting data from both $^1$H and $^{13}$C NMR spectra. DEPT-135 and DEPT 90 $^{13}$C NMR were used to identify the connectivity of the carbon atoms. Also, similarity to the $^{13}$C NMR of the analogous diol, trans-sobrerol, was observed, indicating both compounds have the same carbon backbone.**

**In addition, FTIR was used to confirm organic nitrate functionality and CIMS was used to confirm the molecular weight of the proposed structure (215 g/mol).**

**These results indicate a very high confidence in the molecular assignment of the APN.**

Additionally, the authors report that campholenic aldehyde, pinol, and pinocamphone reaction products were identified via GC-MS methods. Since none of these species are commercially available as standards, it's not clear to me how definitive identification of the reaction products was made.

**Upon closer investigation of the data, campholenic aldehyde and pinocamphone could not be conclusively identified as reaction products. The identification of pinol, however, was accomplished via comparison to a known EI mass spectrum and published retention indices, as described in Supplemental Information Section S2.**

Because neither the reactant nor the products of the main chemical reaction under study have been convincingly identified, I believe that these issues must be addressed in a revised manuscript.

**Thank you for the helpful comments. These issues have been addressed in the updated manuscript.**

Anonymous Referee #2

This work investigated the hydrolysis of a-pinene derived organic nitrate (APN), isopropyl nitrate, and isobutyl nitrate in solutions of varying pH. It was found that the hydrolysis rate constant increases with solution acidity for all compounds studied. The lifetime of APN with respect to hydrolysis increases from 8.3 min to 8.8 hr for solutions with pH = 0.25 and pH = 6.9, respectively. Campholenic aldehyde, pinol, and pinocamphone were identified as hydrolysis products. Theoretical calculations were performed for the hydrolysis mechanisms for isobutyl nitrate using Density Function Theory, and analogous mechanisms were then proposed for the hydrolysis of APN. A unimolecular specific acid-catalyzed mechanism was proposed.

The manuscript is generally well-written. The work will be of interest to the atmospheric research community and will add to the growing recent studies on organic nitrate formation and fates. The results are interesting and have important implications. I have several main comments:

Firstly, the manuscript should provide more context and carefully compare and contrast the results from this study to prior published work. Instead of just briefly mentioning prior literature, the authors should discuss any discrepancies in more detail. For instance, with Jacobs et al. (ACP, 2014) with respect to the specific acid-catalyzed mechanism, with Darer et al. (ES&T, 2011) and Hu et al. (ACP, 2011) with respect to the hydrolysis rates of primary/secondary organic nitrates, etc. Further, the results reported here are quite similar to the work by Bleier and Elrod ( J Phys Chem A, 2013), who investigated the hydrolysis of a-pinene oxide under acidic conditions. However, other than mentioning that campholenic aldehyde is also identified as a major product in Bleier and Elrod, there is no further discussion regarding similarities/differences between their results and those reported in Bleier and Elrod. This should be addressed. Bleier and Elrod have also identified other products (pinol, etc) and proposed reaction mechanisms for the formation of these products. In this regard, it'd be important for the authors to specify clearly what is new in this study.

**For discussion related to specific acid catalysis and Jacobs et al. (2014), see the reply to Detailed Comment 5 posted below**.

**For discussion related to product identification and Bleier and Elrod (2013), see the reply to Detailed Comment 7 posted below**.

**In the very nice work of Darer et al. (2011), it is clearly shown in Figure 3 that the lifetime of the organic nitrate (for a pH of 0.83, according to our estimate) is approximately 1 h, rather than the 4 minutes (0.061 h) that they report. We wonder if this is a misprint. In any case, the following paragraph was added comparing our results to the previous studies, starting on page 5, line 18:**

**"*The hydrolysis lifetimes measured for the APN are similar to those previously reported for tertiary hydroxyl nitrates. While not as short as the 0.019 h reported by Hu et al. (2011), the measured hydrolysis lifetimes of the APN are consistent with the data presented in Fig. 3 of Darer et al. (2011), which we estimate to display a lifetime of approximately 1 h at pH=0.8. However, a true comparison*

*of the measured hydrolysis rates is not possible, as solution pH values were not reported in either Darer et al. (2011) or Hu et al. (2011). Nonetheless, one possible explanation for the shorter lifetimes observed in previous work compared to this study may be related to the highly acidic hydrolysis conditions used in previous experiments, which employed the use of strong acids (i.e. acids with negative pK$_a$ values) ranging up to 2 M (Darer et al., 2011; Hu et al., 2011). If the organic nitrates studied in both Darer et al. (2011) and Hu et al. (2011) also proceeded via specific acid catalysis, a very low pH environment would likely have led to larger hydrolysis rates than those observed in this study."*

Secondly, the authors did not provide specific details on how the hydrolysis products are identified and verified. Since this is an essential part of the manuscript and affects all the discussions on the proposed mechanisms, this info needs to be included in the revised manuscript.

**The hydrolysis product identification was accomplished by comparison to a known EI mass spectrum and published retention indices, as described in Supplemental Information Section S2.**

Lastly, the organics nitrates discussed in this manuscript are from photooxidation under high NOx condition; however, nitrate radical oxidation of BVOC is another important source of organic nitrates (Ng et al., ACPD, 2016), where the relative abundance of primary/secondary/tertiary organic nitrates and their fates can be different from those formed from OH oxidation. When discussing "hydrolysis of organic nitrates" throughout the manuscript, it is important that the authors make it clear regarding the specific origins of the organic nitrates in relevant discussions in the manuscript, and comment that the results can be different for other types of organic nitrates (from NO3 chemistry).

**Since the oxidation of other monoterpenes can lead to a greater distribution of primary/secondary organic nitrates, the terms "α-pinene-derived organic nitrate" and "tertiary" were specified as necessary. In addition, the following was added on page 7, line 13:**

**"*It is important to note that organic nitrates formed from other oxidation processes, such as nitrate radical addition, may produce a larger degree of primary and secondary species that may be more resistant to hydrolysis than the tertiary organic nitrates formed during photooxidation of substituted alkenes (Nah et al., 2016).*"**

I recommend publication of the manuscript after the authors address the comments.

Detailed Comments

1. Page 2, line 23. I do not think the authors need to specify "under dry conditions", since organic nitrates are typically detected and can comprise a large fraction of ambient SOA mass even under humid ambient and lab conditions.

**The phrase in question was removed. Thank you.**

2. Page 2, line 26. For hydrolysis of organic nitrates, here, it would be appropriate to cite Darer et al., (2011), Hu et al., (2011), and Boyd et al., (2015) as well.

**The Boyd et al. (2015) reference was added. Darer et al. (2011) and Hu et al. (2011) were not added here because this line is referring to hydrolysis occurring in the aerosol phase rather than in bulk solution.**

3. Page 5, line 8. The specific acid-catalyzed mechanism proposed here is in contrast to that in Jacobs et al. The authors noted that in Jacobs et al. the solution pH was not reported. The authors are correct that the pH values were not reported, however, according Jacobs et al, "The hydrolysis of the hydroxynitrate isomers was monitored in a variety of acid concentrations (0 to 2 M HClO4), and, regardless of the acid concentration, the rate of hydrolysis remained constant". With this, it would seem that the range of pH values in Jacobs et al. is comparable to this study? If so, the authors need to evaluate and discuss this discrepancy more extensively.

**The reaction rate of general acid catalysis is dependent on buffer concentration while specific acid catalysis rates do not depend on buffer concentration. Thus, the following was added on page 5, line 5:**

**"*It is important to note, however, that Jacobs et al. (2014) did not observe an increase in hydrolysis rates with increasing buffer concentration, a result that is a defining characteristic of specific acid catalysis.*"**

4. Page 5, line 9. The authors noted that essentially identical kinetics were observed for primary and secondary nitrates, which in in contrast to Darer et al. and Hu et al. where hydrolysis rates increase with alkyl substitution. What is the cause of this discrepancy? Because of different mechanisms? Please discuss in more detail.

**The observed discrepancy is related to the stability of the carbocation intermediates (see second paragraph on page 5). In addition, the following was added on page 5, line 14:**

**"*Since carbocation stability drives the rate of the unimolecular hydrolysis reaction, and rearrangements were not possible for the structures studied in Darer et al. (2011) and Hu et al. (2011), the observed hydrolysis rates in the previous studies followed a trend based on the degree of alkyl substitution, as opposed to this work and Jacobs et al. (2014).*"**

5. Page 6, line 8. The authors noted that campholenic aldehyde, pinol, and pinocamphone were identified as major products.
a. How is product identification verified? Did the authors synthesize authentic compounds of these products? Please clarify.

**The hydrolysis product identification was accomplished by comparison to a known EI-MS spectrum and published retention indices, as described in Supplemental Information Section S2. It is important to note that upon closer investigation of the data, campholenic aldehyde and pinocamphone could not be conclusively identified as reaction products.**

b. As samples at different time points were analyzed (page 3 line 25), please include time dependent data for these products in the revised manuscript, which can offer insights into the dynamics of the formation of these species and/or potential further reactions in the particle phase.

**Unfortunately, only the reaction endpoints were available for GC-MS analysis. To clarify, the following was added on page 3, line 27:**

**_"GC-MS data, which was used for product identification (see Section S2 in the Supplemental Information), was only available for data collected at the reaction endpoints."_**

6. Page 7, first paragraph. The authors wrote ". . .partitioning of atmospheric organic nitrates to aerosol particles is not likely to induce further reactions capable of releasing gas phase NOx to the atmosphere. . .." Other than hydrolysis, organic nitrates can also be photolyzed and/or react with OH further, and depending on the VOC, they can also release NOx back to the atmosphere upon photooxidation (e.g., Nah et al., ES&T, 2016). The authors should modify the text and take both hydrolysis and photooxidation into account when discussing organic nitrates fates and implications of their work.

**Thank you for the comment. Section 4 (Discussion) was amended to include detailed discussion on the fate of organic nitrates. Starting on page 7, line 6:**

**_"The reported hydrolysis lifetimes for atmospherically-relevant hydroxyl nitrates are generally shorter than those reported for oxidation reactions, which range from 2.5-28 h for gas phase OH radical oxidation and from 0.7-28 h for gas phase $O_3$ oxidation (Lee et al., 2014). Gas phase photolysis lifetimes have been reported from 0.50-1.0 h for carbonyl nitrates (Müller et al., 2014; Xiong et al., 2016) to ~107 h for mono-functional organic nitrates (Higgins et al., 2014). In comparison, previously reported tertiary organic nitrate hydrolysis lifetimes have been measured as short as 0.019 h (Hu et al., 2011), and as short as 0.14 h under acidic conditions in this study. This indicates that the partitioning of organic nitrates to the particle phase is likely a significant sink for atmospheric $NO_x$ that may critically diminish the potential for long range transport of $NO_x$/ozone in the form of organic nitrates. It is important to note that organic nitrates formed from other oxidation processes, such as nitrate radical addition, may produce a larger degree of primary and secondary species that may be more resistant to hydrolysis than the tertiary organic nitrates formed during photooxidation of substituted alkenes (Nah et al., 2016)._**

**_To our knowledge, there are currently no reported photolysis or oxidation rate constant measurements for α-pinene-derived organic nitrates. However, using the EPI Suite available at the Environmental Protection Agency website (http://www.epa.gov/opptintr/exposure/pubs/episuite.htm), gas phase atmospheric lifetimes of APN were calculated to be 1.3 h and 0.64 h for OH radical and $O_3$ induced oxidation, respectively. Due to the lack of carbonyl functionality, it is expected that the APN photolysis rate is negligibly small. With recently reported aerosol pH values ranging from pH 0.5 to 3.0 in the southeastern US (Guo et al., 2015), the corresponding ambient hydrolysis lifetimes of the APN would be on the order of a half hour, which would indicate that particle phase hydrolysis is the principal atmospheric sink for α-pinene-derived organic nitrate compounds, in comparison to removal via photolysis and oxidation pathways. Hydrolysis in chamber experiments may be even faster as aerosol pH has recently been measured as low as pH -0.68 for laboratory-generated particles (Rindelaub et al., 2016)."_**

7. Page 7, line 8. The author wrote ". . .Campholenic aldehyde has also been identified as the major product of the hydrolysis of another _-pinene oxidation product, _-pinene oxide (Bleier and Elrod, 2013)..".
a. Other than campholenic aldehyde, just like this study, Bleier and Elrod also identified pinol as a product in their study. Bleier and Elrod also proposed reaction mechanisms for the formation of campholenic aldehyde and pinol. The authors should also mention these results from Bleier and Elrod in their manuscript.

**Further data analysis has concluded that campholenic aldehyde cannot be conclusively identified as a product of this study. The following was added regarding pinol formation on page 6, line 27:**

**"*Pinol has also been observed as a product of α-pinene oxide hydrolysis (Bleier and Elrod, 2013), which is expected to proceed via the same tertiary carbocation intermediate as the APN hydrolysis.*"**

b. Bleier and Elrod identified trans-carveol and trans-sobrerol as hydrolysis products in their study. It did not appear that these products are identified in this study? What is the cause of this discrepancy? The experiments in Bleier and Elrod were also conducted under acidic conditions. c. Overall, as this work is quite similar to that by Bleier and Elrod, the authors should compare and contrast their results to those from Bleier and Elrod in more detail, e.g., products, formation mechanisms, etc, and specifically clarify what is new in this study.

**Neither carveol nor sobrerol were observed in this study, which may be related to the relative thermodynamic stability of the products. As such, the following was added on page 6, line 28:**

**"*In contrast to this study, Bleier and Elrod (2013) also observed the formation of sobrerol and carveol, the alternative $S_N1$ and E1 products, respectively, derived the tertiary carbocation. While it is peculiar that neither sobrerol nor carveol were detected in this study, the differentiation can at least partially be explained by the greater thermodynamic stability of pinol in comparison to both sobrerol and carveol (Bleier and Elrod, 2013).*"**

8. Page 7, line 13, I think the authors should make it clear that this is the case of organic nitrates formed from photooxidation. For organic nitrates formed from nitrate radical oxidation, the relative amount of primary/secondary/tertiary nitrates and their overall hydrolysis could be different (Boyd et al., ACP, 2015).

**Thank you for the comment. This line was replaced with the following on page 7, line 13:**

**"*It is important to note that organic nitrates formed from other oxidation processes, such as nitrate radical addition, may produce a larger degree of primary and secondary species that may be more resistant to hydrolysis than the tertiary organic nitrates formed during photooxidation (Nah et al., 2016).*"**

9. Figure 1 and Figure 6. Which organic nitrate did the authors synthesize?

**Thank you for the observation. Figure 1 has been updated.**

10. Figure 6. The chemical structure for pinol is incorrect.

**Thank you for the observation. Figure 6 has been updated.**

[revised manuscript text omitted]